# Recent Advances in Touch Sensors for Flexible Wearable Devices

**DOI:** 10.3390/s22124460

**Published:** 2022-06-13

**Authors:** Abdul Hakeem Anwer, Nishat Khan, Mohd Zahid Ansari, Sang-Soo Baek, Hoon Yi, Soeun Kim, Seung Man Noh, Changyoon Jeong

**Affiliations:** 1School of Mechanical Engineering, Yeungnam University, 280 Daehak-Ro, Gyeongsan 38541, Korea; hakeemanwer1@gmail.com; 2Industrial Chemistry Research Laboratory, Department of Chemistry, Faculty of Sciences, Aligarh Muslim University, Aligarh 202 002, India; nishatk57@gmail.com; 3School of Materials Science and Engineering, Yeungnam University, 280 Daehak-Ro, Gyeongsan 38541, Korea; zahid.smr@yu.ac.kr; 4Department of Environmental Engineering, Yeungnam University, Gyeongsan 38541, Korea; ssbaek@yu.ac.kr; 5Mechanical Technology Group, Global Manufacturing Center, Samsung Electro-Mechanics Co., 150 Maeyeong-ro, Yeongtong-gu, Suwon 16674, Korea; poemisty@gmail.com; 6Research Center for Green Fine Chemicals, Korea Research Institute of Chemical Technology, Ulsan 44412, Korea; soeun12@krict.re.kr

**Keywords:** flexible sensor, wearable devices, touch sensor, nanocomposite, resistive touch sensor, piezoelectric touch sensor, triboelectric touch sensor

## Abstract

Many modern user interfaces are based on touch, and such sensors are widely used in displays, Internet of Things (IoT) projects, and robotics. From lamps to touchscreens of smartphones, these user interfaces can be found in an array of applications. However, traditional touch sensors are bulky, complicated, inflexible, and difficult-to-wear devices made of stiff materials. The touch screen is gaining further importance with the trend of current IoT technology flexibly and comfortably used on the skin or clothing to affect different aspects of human life. This review presents an updated overview of the recent advances in this area. Exciting advances in various aspects of touch sensing are discussed, with particular focus on materials, manufacturing, enhancements, and applications of flexible wearable sensors. This review further elaborates on the theoretical principles of various types of touch sensors, including resistive, piezoelectric, and capacitive sensors. The traditional and novel hybrid materials and manufacturing technologies of flexible sensors are considered. This review highlights the multidisciplinary applications of flexible touch sensors, such as e-textiles, e-skins, e-control, and e-healthcare. Finally, the obstacles and prospects for future research that are critical to the broader development and adoption of the technology are surveyed.

## 1. Introduction: Development of Wearable Devices

During the past decade, advances in artificial intelligence (AI), sensors, and communication technologies have dramatically improved wearable sensors. The research and development of wearable sensing technology are accelerating the development of new applications in entertainment, fitness, and gaming, and specialized applications in certain fields, such as defense, healthcare, and security. Wearable devices had an estimated market value of USD 80 billion in 2020, representing a threefold increase since 2014; their market value is predicted to reach USD 138 billion by 2025 [1]. Smartwatches and wristbands had a combined market of 51% among the wearables used by consumers in 2019, and ear-worn wearable devices were expected to have a market share of 48% by 2021. In contrast, wristbands and smartwatches were expected to have a combined market share of 37%. The best-known wearables are ear-worn devices, with a market share of 48% in 2019, followed by smartwatches and wristbands with a combined share of 37% in 2020. Ear-worn devices will dominate the market with 48%, followed by smartwatches and wristbands with a combined share of 37% (Figure 1) [2]. 

In the past few years, ear-worn wearables—in particular, true wireless stereo wearables—have exploded from a very small share to a large share of the wearable device market following the introduction of Apple AirPods in 2016; their popularity has also exponentially increased during the current coronavirus disease (COVID-19) pandemic [2] caused by severe acute respiratory syndrome coronavirus 2 (SARS-CoV-2), because many people worldwide have been studying and working from their residences. The pandemic has led to an active interest in researching and developing reusable smart masks that can detect SARS-CoV-2 and self-sterilize [3,4]. In addition to the COVID-19 pandemic, other consumer wearable technologies have also been affected in fields, such as contact tracing, patient tracking, remote patient monitoring, and mobile payment systems [5,6]. The market for wearable sensing technologies was dominated by medical/fitness connected services in 2020. The wearable market is also booming in other segments, such as entertainment or gaming, industrial wearables, defense, and security (Table 1). Approximately USD 10 billion of market growth is expected by 2025 for the wearable payments services market (approximately USD 72 billion by 2025), compared with the combined fitness and medical wearables market [2]. Near-field communication (NFC) is one of the technologies that have expanded the wearable payments market. Financial payment standards are being supported in smartphones, and NFC will be incorporated into smartwatches, fitness bands, and other wearables in the near future [7].

Wearables devices for human health have received intensive interest because they are an integral part of AI and the IoT, and commercial wearable devices that monitor human body signals, such as the Mi Band, Apple Watch, and many others, continue to be introduced [2]. Conventional sensors, which are usually made of rigid materials (metals or semiconductors), are rigid, inflexible, and difficult to wear. By contrast, flexible sensors have advantages over conventional rigid devices. These sensors are nontoxic and lightweight and can be worn comfortably on the body [8,9].

Researchers worldwide have recently begun to turn their attention to flexible touch sensors as a component of flexible wearable sensors. Several investigations involving using flexible sensors to monitor the health of patients [10,11] and detect human motion [12,13], as well as investigations in which flexible sensors were incorporated into electronic skins [14,15] have been reported.

This brief review highlights the latest advances in flexible sensors by emphasizing sensors integrated into garments or worn directly on the skin for diverse applications. Emphasis is placed on developments reported within the past few years, as described in Figure 2. Section 1 and Section 2 of this review introduce the basic principles of various touch sensors, including those based on resistive touch, capacitive touch, triboelectric touch, and piezoelectric touch. Section 3 discusses sensing materials, emphasizing some specific nanomaterials with numerous advantages. Section 4 discusses manufacturing technologies with critical methodologies currently being researched. Several practical approaches are discussed in Section 5. The topics covered in Section 5 are the most important for updating the research fields and uses for flexible touch sensors in everyday life. In view of these advances, Section 6 discusses the next steps in developing sensors both in the laboratory and in industrial manufacturing.

## 2. Touch-Sensor Working Principles

There are three types of sensing actions: touch, movement, and pressure. The distance over which actions are performed differentiates fingertip touch and movement, and fingertip pressure level provides a third sensing action. Regular touch occurs when the finger is at a zero distance from the sensors and no force is applied. With increasing pressure, the amount of force increases (pressure sensing). Sensors that detect movement can use a combination of multiple sensing points or proximity sensors (the current distance exceeds zero). This review explores touch motions, where the separation between the finger and sensors is zero, and the pressure is less than 10 kPa. Table 2 summarizes several types of touch sensors that are compared on the basis of their materials, electrical or mechanical performance, working principles, and sensor thickness. 

### 2.1. Single Mechanism

#### 2.1.1. Capacitive Touch Sensors

Capacitive sensor technologies can be broadly divided into two main categories: mutual-capacitive and self-capacitive technologies [16]. For a self-capacitive system, capacitive changes are measured relative to earth ground. It operates on the basis of the parallel-layer model, where the electrode and bottom (or user’s finger) act as the two layers of the capacitor. As capacitance is added with each “touch”, the capacitance on the electrode of the self-capacitive system increases [26]. Contrarily, any intended or unintended capacitance between two “charge-holding objects” can be termed mutual capacitance [27]. A mutual capacitance between intersecting elements of columns and rows can be intentionally created via projected capacitance touch sensors. With the help of such system electronics, a single scan can detect multiple touches by measuring each node (intersection) individually. The surface capacitance measurement is used by some touch devices to determine the human body (finger). The sensor is contacted by a finger, forming a capacitor dynamically [28]. 

With robust and mechanically simple capacitive elements, such capacitive touch sensors can function over a wide range of pressures and temperatures. These capacitors are inherently low power, exhibit small hysteresis, and are suitable for wireless applications because of the lack of direct current (DC) flowing through the capacitor. However, in the vicinity of electronic devices, such sensors have disadvantages stemming from their nonlinearity and stray capacitance.

#### 2.1.2. Resistive Touch Sensors 

Resistive touch sensors respond to pressure [29]. These sensors are composed of several layers, with the most critical layer formed by two electrodes separated by another layer with poor conduction. Herein, a light pressure resulting from contact can predominantly change the resistance. Apart from a finger, other stimuli, such as a stylus, can also be sensed by these sensors. Resistive touch sensors were the earliest used sensors, with the advantages of low cost, fast response, a linear output, and high durability. However, such sensors require power, making them unsuitable for wearable systems (i.e., low-power systems) [12]. 

Piezoresistive materials, such as metals or semiconductors, can be used in these sensors. Upon application of mechanical stress, these materials exhibit a change in electrical resistance (i.e., a change in the distance among charged particles). The performance of such sensors can be dramatically improved by the implementation of geometric microstructure designs, such as pillars, pyramids, and hemispheres [19]. Because of the nonuniformity of the stress distribution, the pyramid structure is the most widely used microstructure. The tips of pyramid structures tend to exert more pressure than other structures because of their more significant mechanical deformation and sensitivity [30,31].

#### 2.1.3. Piezoelectric and Triboelectric Touch Sensors 

Piezoelectric materials translate deformation into electrical energy, enabling the development of a piezoelectric sensor [22]. Applying slight mechanical pressure (light touch) to a so-called piezoelectric material results in the separation of the electric charges because of electrical dipole moments, generating electrical voltage. When the external force is withdrawn, the polarization phenomenon disappears. By detecting the change of the electrical signal, it can be used to realize pressure sensing. Pressure sensing can be achieved by detecting the changing electrical signal. Some of the commonly used piezoelectric materials are BaTiO_3_, PZT, ZnO, PVDF, PbTiO_3_, and polypropylene (PP) [31]. Piezoelectric sensor performance may hinder as a result of cracking due to the pressure applied to traditional piezoelectric materials to overcome the problem of cracks and hole defects. Yang et al. synthesized a PDA@BTO/PVDF composite by introducing polydopamine (PDA) as a surface modification agent to barium titanate (BaTiO_3_, BTO), followed by their blending with a poly(vinylidene fluoride) (PVDF) matrix [32]. Piezoelectric sensors are commonly used for detecting various human body signals, including respiratory patterns, pulse signals, or recording finger movements [33], and to obtain heart rate signals [33].

#### 2.1.4. Triboelectric Touch Sensors

Triboelectric sensors (or contact-electrification) is employed by a triboelectric nanogenerator (TENG) [21,34], where a physical contact (a touch) generates a potential electrical signal without reliance on an external power supply. 

Triboelectric nanogenerators (TENGs) form the basis of triboelectric sensors. The inductive charges are generated on the friction layers of TENGs under external forces owing to the electron affinity differences. Upon the removal of the external force, current starts to flow between electrodes as a result of the formation of internal potential. Moreover, the current signals are analyzed for the sensing function. Sensors using electrostatic induction for the detection of human motions have been available since the introduction of triboelectric nanogenerators in 2012 [35]. The triboelectric nanogenerators may transform the received signals of body movements into electrical impulses, allowing the sensor to function without the need for external power. Commonly used as friction layer materials, they can easily lose or gain electrons, such as PTFE, PDMS, PI, nylon [36] copper, and silver. In triboelectric sensors for human health, a range of innovative functional materials has recently been used. Hydrogel, for example, is commonly used for the electrodes in triboelectric sensors capable of detecting arbitrary human body movements, such as deformation and stretch. Bionic structure sensors fabricated using ionic liquids can monitor underwater movements of the human body, as shown by Zou et al. Textile-based wearable sensors, in particular, are the emerging trend in sensing physiological signals [37].

E-textile and e-skin sensors make use of triboelectric and piezoelectric effects. Several studies have been reported to enhance the electric output via circuit design and material aspect, decrease the loss of power due to signal irregularity, or contact surface optimization. However, poor dynamic, stretchability, and low flexibility performance have restricted their practical applications. The sensor elements in such sensors are self-powered, robustness, and low power, which account for their main advantages. However, their significant disadvantages include the complexity and difficulty of integrating into a system.

The presence of several contacts simultaneously can be recognized by a surface using a technology known as multi-touch. This is mainly based on the sensor arrays and allows multiple finger gestures, such as swipe, zoom in/zoom out, scroll, and select. Resistive, capacitive, triboelectric, or optical principles can be used to fabricate these sensor arrays. However, capacitive or resistive sensor arrays have been demonstrated by several studies as the best suited for multi-touch surfaces [25,38,39].

## 3. Hybrid Multisensor Mechanisms

The multifunctional sensing hybrid mechanism is a crucial tool for enhancing the sensitivity of wearable devices. The hybrid mechanism involves integration mechanisms based on two or more principles. However, in this section, hybrid-principle mechanisms for different applications are discussed.

### 3.1. Two-Principle Integration Mechanism

A wearable sensor based on a single-principle mechanism is unable to detect multiple features and could not meet the requirements for multifunctional detection. Therefore, it is necessary to develop hybrid mechanisms to detect multiple properties and enable the fabrication of multifunctional sensors.

Two-principle mechanisms consist of various integrated mechanisms, such as triboelectric–piezoelectric mechanisms or capacitive–piezoelectric mechanisms. These two-principle mechanisms are used extensively in wearable devices. Tang et al. demonstrated a working mechanism based on simultaneous triboelectric and piezoelectric mechanisms in self-generated powered sensors [40]. Here, the two-principle mechanisms operate in different directions because the triboelectric components consist of four working electrodes that operate in the *xy* plane, whereas the piezoelectric element functions in the *z* plane. The mechanism’s structure is shown in Figure 3a. This integration has shown extensive applicability in the human–machine interaction and other wearable electronic devices [40].

Another component based on triboelectric and resistive integration mechanisms was investigated to detect gestures of prosthesis motion or robotic motion and of the human body [33]. Static and dynamic gestures are detected on the basis of differences in the activity of these integrative principles. In the case of static gestures, the response is measured in terms of the area of the conductive layer. The reduction in the area of the conductive layer leads to an increase in the resistance. By contrast, in the case of detecting dynamic motions, the triboelectric component functions as a nanogenerator to help sense different movements. Such devices could be used as a smart skin to perform and track other activities of human body parts. With the development of new materials, other authors have reported similar works in the field of wearable devices. Chen et al. [41] developed carbon nanotube-based poly-dimethylsiloxane (CNT-PDMS) and used it as a working electrode material in devices to track fingers. They used different types of CNT-PDMS (double-spiral and porous) as electrodes, and the substrates were fingerprints, epidermis, and dermis; thus, the CNT-PDMS could ultimately be used to create a whole set of e-skin sensors. As shown in Figure 3b, the mechanism is based on the change in voltage due to the triboelectrification effect, which is used to sense sliding motion. Additionally, pressure sensing is possible via a change in resistance [41].

Ma et al. [42] also reported using PDMS in conjunction with carbon black (CB)/thermoplastic polyurethane (TPU). Their hybrid device was a triboelectric–piezoresistive sensing unit that can easily sense and identify the mode of different boxing punches.

Park et al. [43] proposed using a combination of piezoelectric and resistive hybrid mechanisms for creating smart skin. The materials used were polyvinylidene difluoride (PVDF) and reduced graphene oxide (r-GO) microdome structures. The hybrid system of PVDF–rGO exhibited both piezoelectric and resistive properties, enabling it to easily perceive static and dynamic pressures and temperatures [43]. Extensive research on hybrid integration mechanisms involving capacitive and resistive modes has been reported in the literature. Chen et al. [44] used a Taiwan Semiconductor Manufacturing Company (TSMC) standard complementary metal-oxide semiconductor (CMOS) process to develop a tactile force sensor that consists of capacitive membranes and piezoresistive bridges that can sense force in the vertical direction and shear force, respectively. Park et al. [45] developed the first e-skin that can detect pressure, strain, bending, and lateral strain simultaneously. This developed material can also harvest energy using external stimuli, leading to self-powered devices and maximizing its applicability in different wearable and other devices. The architecture of the proposed sensor is shown in Figure 3c.

Moreover, the integrative aspects of capacitive and piezoresistive effects are also beneficial in creating a flexible sensor that can detect both touch and pressure, as reported by Hwang et al. [46]. Figure 3d shows a sandwich structure in which a change in capacitance induced by a change in the current would ultimately be beneficial in detecting external touch. By contrast, a change in the piezo resistance of the electrode induced by a decrease in the current would enable the detection of pressure. Because of these important properties, these materials can be extensively integrated into stretchable touch panels and other wearable devices [46].

Zhu et al. reported a thermoelectric and piezoelectric mechanism based on composites of organic and thermoelectric materials, such as PVDF and polyaniline, respectively [47]. These composites were used in flexible composite-based sensors that could simultaneously detect tactile motion and temperature independently of one another. These sensors exhibited vastly enhanced sensing of various motions. Because the piezoelectric materials also possess some pyroelectric properties, various authors have reported using multifunctional sensors that exhibit these properties. Chen et al. demonstrated a polymeric-substrate-based flexible sensor that can easily sense temperature and acoustic signals [48].

Thus, hybrid sensors with different sensing mechanisms provide excellent opportunities for miniaturization and multifunctionality. The further exploration of these aspects will be a focus of future research.

**Figure 3 sensors-22-04460-f003:**
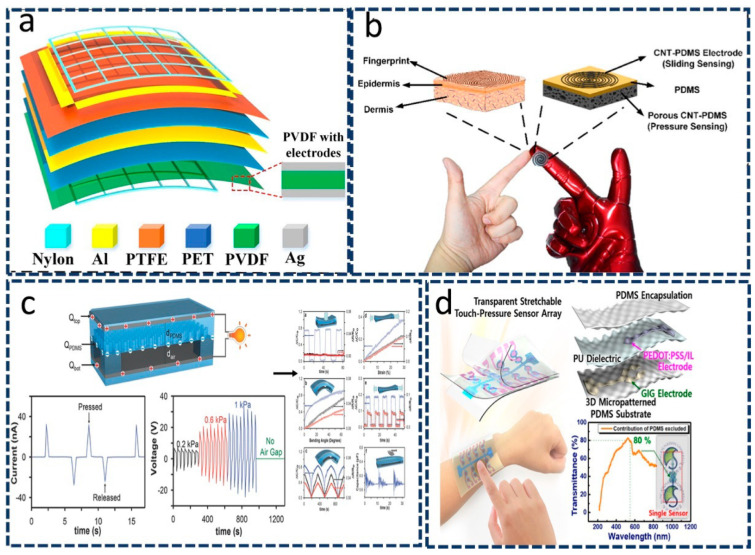
Two−principle integrated wearable sensors. (**a**) An integrated triboelectric–piezoelectric self-powered sensor. Reproduced with permission from Elsevier (2021) [40]. (**b**) A comparison diagram of fingertip and e-skin. Reproduced with permission from Elsevier (2017) [41]. (**c**) The e-skin can not only sense strain, pressure, and bending, but also detect lateral strain. Reproduced with permission from Wiley (2014) [45]. (**d**) Through the capacitive and piezoresistive effects of the transparent flexible sensor, touch and pressure can be detected. Reproduced with permission from Springer Nature (2019) [46].

### 3.2. Three-Principle Integration

Hybridization involving three integration mechanisms will play a tremendous role in future wearable devices for use in wide applications, such as the human–machine interface, smart robots, and wearable electronics. To further improve the performance of sensing materials, it is important to introduce different mechanistic parameters into sensing mechanisms to expand their applicability in wearable and other devices. Various polymeric materials, such as PVDF, PDMS, PET, polyurethane (PU), polypropylene (PP), and other nanomaterials, are required to improve the cracking performance of materials subjected to great pressure or temperature changes, thereby requiring the integration of more than one mechanism for enhancing the sensing ability of the developed materials. Zhao et al. [49], for example, developed high-performance sensors by combining three effects: triboelectricity, piezoresistivity, and piezoelectricity. They fabricated a nanofilm of carbonized polyacrylonitrile/barium titanate (PAN-C/BTO) using the electrospinning method (Figure 4a). The resultant nanofilms could independently and simultaneously detect pressure and curvature with high and enhanced sensitivity. These sensors could be used multifariously in detecting human motion, such as finger tapping and swallowing. Kim et al. [50] reported using capacitive, resistive, and triboelectric effects to generate a self-powered sensor. Figure 4b clarifies that a change in capacitance can be used to enable the sensing of vertical force, whereas a change in resistance enables the measurement of lateral strain. As these changes occur, silver yarns inside the wall rub against each other, which generates energy. The three integrative mechanisms, along with the ability to self-generate power, will be extremely useful in developing portable and wearable devices [50].

Triple-principle mechanisms involving piezoelectric, triboelectric, and pyroelectric effects have been reported by Wang et al. [51]. These three mechanisms led to the development of a transparent triple hybrid nanogenerator. Figure 4c shows PVDF nanowires in a PDMS film used as a triboelectric layer, along with a piezoelectric and pyroelectric layer developed using polarized PVDF and indium tin oxide-based electrodes, respectively [51]. These generators produce various forms of energy that can be efficiently used to operate multifunctional wearable sensors [33]. In summary, the hybrid mechanism involving the core working structure of wearable devices plays an important role in imparting multifunctionality and in miniaturizing wearable devices, enabling the development of accurate and highly reliable sensors in wearable devices and other healthcare detection kits.

## 4. Nanocomposite Material for Flexible Touch Sensors

Flexible sensors should be made of a lightweight material that is biocompatible, comfortable, and does not cause irritation. Nanocomposite materials mostly include metallic thin films [52,53], carbon nanotubes (CNTs) [54,55], metal nanowires (NWs) [56], metal nanoparticles (NPs) [57], and conductive polymers [58,59]. Good electrochemical activity, high electrical conductivity, and a large active area make the NWs, CNTs, and conductive polymers preferred choices for sensors. In addition, NW/CNT composites can be coated/printed directly onto a substrate [60] to produce a sensor with high sensitivity, stretchability, and durability. Conductive polymers, such as poly(3,4-ethylene dioxythiophene) (PEDOT), and especially its complex with poly(styrene sulfonate) (PEDOT:PSS), can be synthesized by chemical or electrochemical deposition [61] and exhibit high conductivity, good transmission of light, good processability in water, and high flexibility. For the mass production of sensors, printable conductive materials are particularly advantageous because they enable nearly all sensor parts to be printed. Several metal conductive inks have been shown to be promising because of their tendency to disperse in solvents and their compatibility with different printing technologies. Ag nanoparticle-based inks and nanowires have been extensively studied as flexible electrodes or conductors for polyethylene (PE) [61]. However, Cu nanoparticle-based ink, which is inexpensive and highly conductive, has attracted particular attention [62]. Inks based on carbon nanomaterials (e.g., CNTs and graphene) have also been shown to be printable and stretchable for flexible sensors [63,64]. The integration of Ag nanowires (AgNWs) [65], ITO [66], graphene [67], PDMS [68], CNTs [68], and PEDOT:PSS [69] has led to transparent stretchable electrodes suitable for certain touch displays and photovoltaic applications. The fabrication of transparent electrodes/conductors faces a major challenge in the trade-off relationship among flexibility, transparency, and conductivity, which depends on the conductive filler concentration [70]. This challenge is particularly relevant to touch sensors such as touch screens and fingerprint sensors.

Liquid metals and liquid ionic materials are intrinsically flexible. Ionic additives can improve the conductivity and stretchability of PEDOT:PSS, leading to conductivity as high as 4100 S cm^−1^ at 100% strain, as demonstrated by Wang et al. [58], and enabling the fabrication of soft sensors that can detect both positive and negative pressures from −60 to 20 kPa [71]. Liquid metals—specifically, eutectic gallium–indium alloys—can be used to fabricate flexible circuits via the integration of room-temperature liquid metals (RTLMs) and water-soluble poly(vinyl alcohol) (PVA) because of their high conductivity, intrinsic stretchability, and low piezoresistivity. Thus, RTLMs and PVA can be integrated into flexible circuits [72], asymmetric force sensors can be prepared from hydrophilic polymer networks [73], and soft sensors can be produced by a 3D-printed rigid micro-bump [74,75]. In dynamic applications, liquid components introduce reliability problems. An excellent solution would be to embed the liquid metals into elastomers, such as PVA.

Furthermore, a polymer matrix combined with a wide range of nanofillers can produce various polymer nanocomposites. Nanofillers can have either two- or three-dimensional structures, similar to CNTs or graphene. The properties of CNTs, such as their high mechanical strength, high aspect ratio, and outstanding electrical properties, make them a particularly promising material. CNTs can be embedded in various polymer matrices to produce materials with diverse properties. Unfortunately, the high Van der Waals forces of CNTs cause the particles to tend to cluster or agglomerations. Thus, the distribution of CNTs in polymers strongly affects the performance of nanocomposites. There have been many efforts to achieve a homogeneously dispersed CNTs within the polymer matrix. This is required to fabrication sensors and establish the foundation for high repeatability. Numerous approaches have been developed to tailor and improve the dispersion of CNTs/polymer composites via solution mixing, including direct mixing, in situ polymerization and melt processing (Table 3). For example, a core-sheath fiber (CSF) for wearable strain sensors was developed by Tang et al. [76]. The fiber is formed by co-extrusion utilizing Ecoglex (a silicon elastomer) that contains 2 wt % MWCNT; the sheaths are made of neat Ecoflex. The sensor is highly stretchable, up to 600% stretching. They have exhibited different GFs at various stretching regions, from GF = −0.063 (about 25%) to GF = 1378 at 330% stretching. For higher concentration of CNTs (3%) the strain sensor displayed GF = −0.45 (110% strain) and reached GF = 153 (60% stretch). When the sensors were subjected to a quasi-transient step test (with strain up to 100%), overshooting occurred during accelerating and then a relaxation followed.

## 5. Manufacturing Technologies

In current studies, spinning, coating, printing, and transferring are indispensable methods that strongly influence the electromechanical and flexible properties of touch sensors because they govern the material and the manufacturing process used.

Yarns or surfaces can be coated with conductive coatings. In this context, graphene nanoplatelets and carbon nanofibers painted along both sides of a rubber piece result in flexible and stretchable capacitive touch sensors with a low sheet resistance (~10 Ω sq^−1^). Chen et al. [8] developed a flexible touch sensor using electron-induced perpendicular graphene sheets implanted on porous carbon films (Figure 5a). These sheets displayed high sensitivity (0.13 kPa^−1^ at pressures less than 0.1 kPa and 4.41 MPa^−1^ at pressures greater than 10 kPa) as well as a fast response time of 66 ms when the substrate was ~0.5 mm Si.

Conductive printing techniques enable the fabrication of flexible touch sensors using printing technology. A transparent capacitive touch sensor fabricated using the inkjet printing technique has been proposed for use in freestanding nanoparticle arrays made of ultrafine polydopamine with a controlled line-to-line separation (similar to pitch), as illustrated in Figure 5b [86]. A self-powered touch sensor with a polyethylene terephthalate substrate with an area of 10 × 10 mm^2^ and a thickness of 380 µm was found to be capable of powering diodes, supplying power to a device, or charging a capacitor. It is a paper-based harvester screen-printed using a mesh [87]. Recently, researchers have focused extensively on 3D-printing technology because of its high commercialization potential, ease of integration, and large-scale manufacturing capability. Yin et al. [88] developed a new generation of wearable sensors based on polyacrylamide with hydratable salts and polyethylene glycol diacrylate. Their sensors exhibit high sensitivity (0.84 kPa^−1^) in the pressure range from 0.5 to 3 kPa. The electrospinning technique was used to make ultrasmall fibers using electric fields, which is an ideal method for producing soft transparent metal electrodes. For example, a flexible and transparent fingerprint sensor array prepared by electrospinning can detect skin temperature and tactile pressure (Figure 5c) [65]. The polyimide-based substrate used in this method was 25 µm thick. Kweon et al. [89] also demonstrated a polymer-based multifunctional pressure sensor that consisted of a hybrid nanostructure integrated with ultralong metal nanowires and fibers. Specifically, 3D electrospinning and vapor deposition polymerization methods have been used to fabricate conductive shells and core polymer nanofibers composed of poly(vinylidene fluoride-*co*-hexafluoropropylene) and poly(3,4-ethylene dioxythiophene) with a thickness of 1 mm. This sensor operates on the principle of resistive sensing, with a substantial sensitivity of ~13.5 kPa^−1^. In addition, flexible sensors can be fabricated by transferring a conductive material onto textile or silicone. Although this technology is not practical, it serves as a step that can be used to support other technologies [1]. Evaporation is another an important step in the synthesis of thin films of materials used in various flexible devices. The evaporation process typically involves two fundamental processes: evaporation of a heated source material and condensation onto a substrate (Figure 5d). As one example, Vieira et al. [90] proposed a highly heat-sensitive thermoelectric thin-film sensor based on *p*-type SnO*_x_*. They deposited a thermo-oxidation film with a thickness of 60–160 nm at a rate of 2 Å s^−1^ and a pressure of 2 × 10^−5^ mbar under an air atmosphere for 3 h at 250 °C. The resultant SnO*_x_* thin films (60 nm) functioned as a touch sensor with high sensitivity (*V*_signal_ = *V*_noise_ ≈ 20) and a rise time shorter than 1 s.

**Figure 5 sensors-22-04460-f005:**
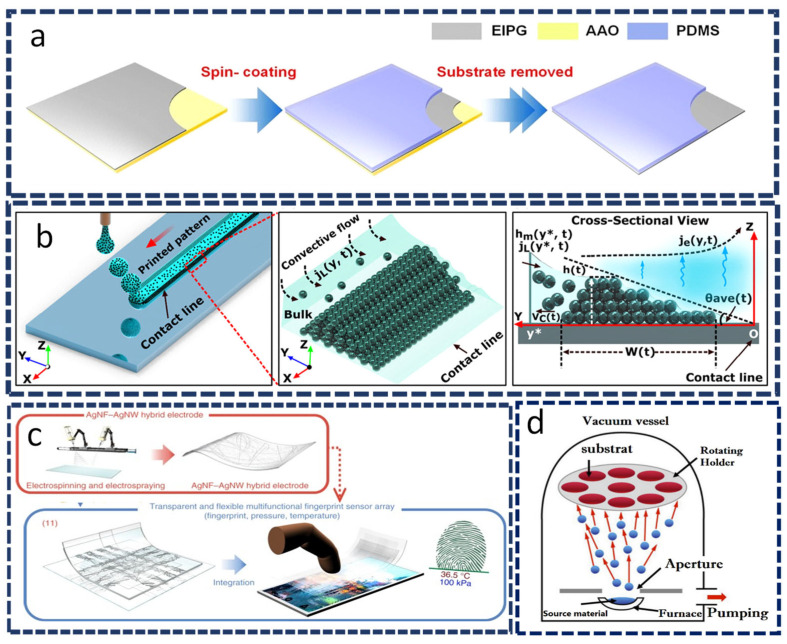
Schematics: (**a**) The flexible electrode is deposited using spin coating with electron-induced perpendicular graphene. Reproduced with permission from Springer Nature (2020) [8]. (**b**) The inkjet-printing controllable process for evaporation-driven convective particle self-assembly at the contact line. Reproduced with permission from Wiley-VCH. (2020) [86]. (**c**) The electrospinning and electrospraying processes of a silver nanofiber (AgNF)–silver nanowire (AgNW) hybrid electrode. Reproduced with permission from Springer Nature (2018) [65]. (**d**) Thermal evaporation coating. Reproduced from Ref. [91].

## 6. Applications of the Touch Types

In a new era, significant advances in the creation of flexible materials for mechanically stretchy and bending sensors will widen the applications of “touch’s definition”. The conductive elements have mechanical and morphological properties that affect not only its intrinsic electrical properties, but also their efficiency and range of applications. Research and advances in sensors have been made over the past few years in the development of e-textiles, e-skins, e-control applications, and e-healthcare. During the last few years, researchers and scientists have made significant advances in developing flexible sensors for e-textiles, e-skins, e-control, and e-healthcare applications.

### 6.1. E-Skins-Based Approach

An electronic skin is an electronic device that mimics human skin properties by being flexible, stretchable, and self-healing. E-skins have multiple applications, including prosthetics, robotics, and skin-attachable devices. E-skins made of ultrathin materials, such as PET or poly(ethylene naphthalate), are a good choice for a small, sustainable deformations. In addition, elastomer substrates, such as polyurethane (PU), latex, and poly(polydimethylsiloxane) (PDMS) may be used in applications that require stretchability [92]. Asghar et al. [93] used magnetically generated microstructures (MPs/PDMSs) to show a piezo-capacitive flexible pressure sensor. In the device, pressure can be sensed over a wide range (0 to 145 kPa) with a fast response (50 ms), as well as high cyclic stability (>9000 cycles). In addition, Park et al. [94] developed a 3D fingertip-shaped artificial skin device with capacitive sensing technology that provides a large amount of electrical signal contact when touched (Figure 6a). It has the ability to detect the precise place of a touch and repair mechanical harm on its own. A combination of ionconductive hydrogel and 3D printing (nozzle size of 0.6 mm) shows an excellent result for touch devices. Naturally, plant materials can also be used as dielectric materials for flexible capacitive e-skins as they are simply composed of dried flower petals or leaves (Figure 6b) [95]. Based on natural-material e-skin (thickness ~207 m), the device was able to operate in a wide range of pressure from 0.6 Pa–115 kPa with a greater sensitivity (1.54 kPa^−1^) and superior strength for over 5000 cycles of pressing or bending. This is a novel strategy for achieving a cost-effective, green and scalable solution.

We also developed a simple method to manufacture e-skins with high efficiency by employing ultrasonic spraying. These e-skins are bioinspired in appearance, consisting of microhexagonal columnar arrays that are tightly interlocked. In our studies, we found that our e-skins with microhexagonal columnar structures were capable of detecting pressure, shear, and bending, as well as static mechanical stimuli. We use piezoresistivity to construct our devices due to surface-to-surface contact between hexagonal structures. SWCNTs were shown to be capable of detecting extremely very small stimuli, as we demonstrated that tightly interlocked hexagonal structures allow direct contact with external stimuli. For example, internal and external minute vibrations as well as tiny water droplets (10 μL) sprinkled on the surface can be detected [96].

### 6.2. E-Textiles-Based Approach

As one of many wearable sensors, e-textiles in stretchy fabrics have drawn attention from both the academic and industrial worlds because of their supreme wearability in allowing seamless fitting across different body sizes and shapes [97]. Traditionally, e-textiles have been created by knitting, weaving, or embroidering functional fibers into fabrics in a twisted or coaxial structure, or embedding functional nanoparticles directly into fabrics. Through adding electronic elements, e-textiles, such as conductive fibers or fabrics, can be used in wearable devices, the human–machine interface, or for controlling/monitoring applications [98].

With the addition of an electronic unit, e-textiles, such as conductive nanofibers or nanofabrics, can be used in wearable devices, data collection, control, and for monitoring applications. A temperature sensor e-textile made up of rGO flakes that have been applied to a bleached cotton yarn used batch dyeing in a high-throughput manner (1000 kg/h) [99]. In addition to the overcoat, cotton yarns were knitted into scaffold shapes via an automatic knitting machine to ensure high mechanical stability against a cyclic analysis at 25–55 °C. The great sensitivity and rapid response time of carbon-based nanomaterials, such as CNTs, have also made them an attractive alternative to rGO flakes for temperature sensors [100]. For example, Hasanpour et al. coated cotton threads with fluorinated ethylene propylene (FEP) and CNTs multiple times following dip-coating processes and drying procedures to protect the temperature sensor from the effects of humidity [101]. A conventional stitching machine was used to stitch threads into polyester fabrics, which conducted cyclic testing without exhibiting significant hysteresis between 50 to 120 °C without the threads unravelling. For even higher sensitivity, Wu et al. developed a hybrid silk fiber structure by mixing CNTs in a 1-ethyl-3-methylimidazolium bis (tri fluoromethyl sulfonyl) imide ionic liquid, which provided additional charge transport paths to enhance sensitivity up to the current highest level of 23.3 kW °C^−1^ [102]. Combining CNT networks with the fast-ionic mobility of an ionic liquid enabled the hybrid structure to improve sensitivity by exploiting the percolation theory of the hybrid structure.

A number of e-textiles have the benefit of being very stretchable, bendable, and washable while maintaining excellent electrical conductivity. In Figure 6c, omniphobic triboelectric nanogenerators (RF-TENGs) embedded into an e-textile provide high performance under deformation, washing, and touch as well as low production costs [103]. E-textiles provide a tremendous benefit of comfort for users due to artificial and natural fibers/fabrics, such as silk, cotton, or polyacrylates, which are common materials in everyday life.

### 6.3. E-Healthcare-Based Approach

For healthcare implementation devices must be low energy consumers and biocompatible to prevent skin irritation. In e-healthcare, wearable bio-chemical/chemical sensors are featured as examples of a flexible touch-sensing. Their greatest issue is maintaining precision in a variable working environment with multiple impacts, such as temperature and humidity [104]. Normally, these biochemical or chemical sensors are often connected to human skin or are woven into garments and fabrics in order to detect external toxins or monitor the level of specific (blood) compounds 7/7 [105]. A near-infrared photoplethysmogram (NIR-PPG) hybrid sensor was proposed for cardiovascular monitoring by Xu et al. [106]. The NIR-PPG is made up of an organic-phototransistor (OPT) with great sensibility and an inorganic light-emitting diode with excellent efficiency. A continuous heart rate variability (real time) can be measured by this sensor using low power (1.2 × 10^−15^ W Hz^−1/2^). Because of the thin-encapsulation structure, the device is highly flexible and can be transferred directly to the finger (Figure 6d). Similarly, the temperature of the tissue at the site of the wound can also be sensed by using a breathable electronic device [107]. A nanomeshed thermoresponsive polymer film with a conductive network is crosslinked by electrospun moxifloxacin-hydrochloride loaded on the device shown in a Figure 6e. Touch-area dimensions are 3 × 3.5 cm^2^ and thickness (~500 μm). In addition, when a wound becomes infected, this nanomesh layer can operate as a very effective heater, which stimulates the release of antibiotics and inhibits bacterial colonization.
Figure 6(**a**) 3D-printed fingertip-shaped artificial skin device that senses precise touch location and automatically heals mechanical damage. Reproduced with permission from ACS (2020) [94]. (**b**) An illustration of an e-skin, which consists of two electrodes with a layer of dielectric material (flower and leaf) between them. Reproduced with permission from Wiley-VCH (2018) [95]. (**c**) Triboelectric nanogenerators (RF-TENGs) fabricated from e-textiles shaped similar to cats that power two LEDs embroidered on the body (when touched). Reproduced with permission from Wiley-VCH (2019) [103]. (**d**) It is comprised of an OPT and an LED that are laminated directly onto the finger to form a PPG sensor for the purposes of cardiovascular monitoring. Reproduced with permission from Wiley-VCH (2017) [106]. (**e**) A finger is attached to the nanomesh film devices for tissue temperature sensing. Reproduced with permission from Wiley-VCH (2019) [107].
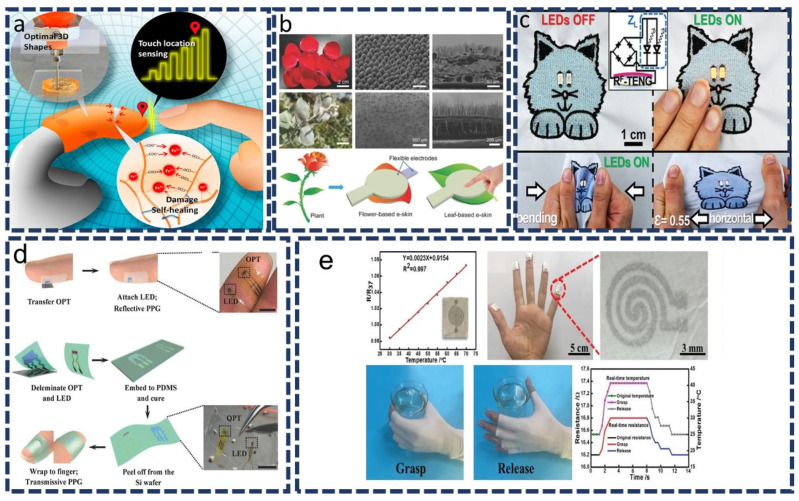


## 7. Summary and Outlook

We presented an update on the latest research in the field of flexible wearable sensors with an emphasis on touch-sensing. As discussed, the capabilities and concepts of the technology for touch sensors have expanded with the development of new materials. The areas of e-skins, e-textiles, e-healthcare, and e-control are key fields with considerable potential.

Future works and commercialization efforts face numerous hurdles. Three critical issues are materials, technology, and the working environment. For instance, metal nanomaterials that are extensively being used in sensor design show relatively poor long-term stability, can be easily oxidized or desulfurized, and are genotoxic. However, CNTs and graphene can have immunological effects, cause lung inflammation, interstitial fibrosis, or asthma, and are possible carcinogens [108]. The encapsulation of sensors is a potential solution; however, sensor performance is affected by encapsulation. The encapsulation (by silicone or PDMS) increases the thickness, resulting in a decrease in flexibility and an increase in the touch pressure required to induce a change inside the sensors. In addition, integrating flexible sensors and electronic devices or systems is difficult because a power supply is required to support a fully functional sensing system. Efforts are underway to enhance the integration capability and flexibility of sensors in such systems. The signal-processing circuit poses another hurdle because the encapsulation approach can result in undesired noise or hotspots when the sensor is in contact with human skin [109]. Moreover, a false trigger can be caused by the presence of uncontrolled working environments, such as vapor, noise, and oils/sweat from the human body. Touch-sensing requires an optimization of the techniques used to collect or analyze data, such as distinguishable intended and false touches. The aforementioned problems necessitate multidisciplinary research to develop solutions, with the goal of designing a complete touch-sensing system and bringing items from the laboratory to the industry.

## Figures and Tables

**Figure 1 sensors-22-04460-f001:**
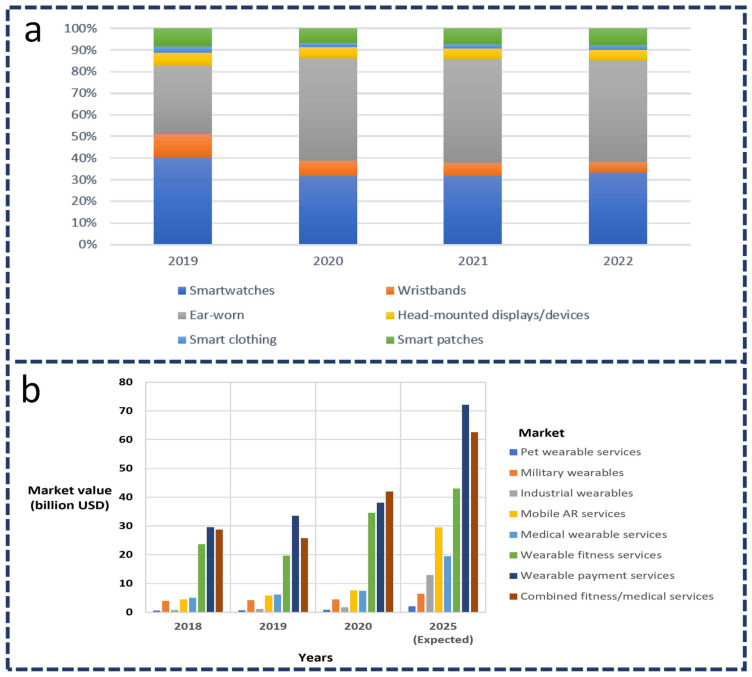
(**a**) Global market share of wearables devices for consumers from 2019–2022; (**b**) wearable services market-value images. Reproduced from Ref. [2].

**Figure 2 sensors-22-04460-f002:**
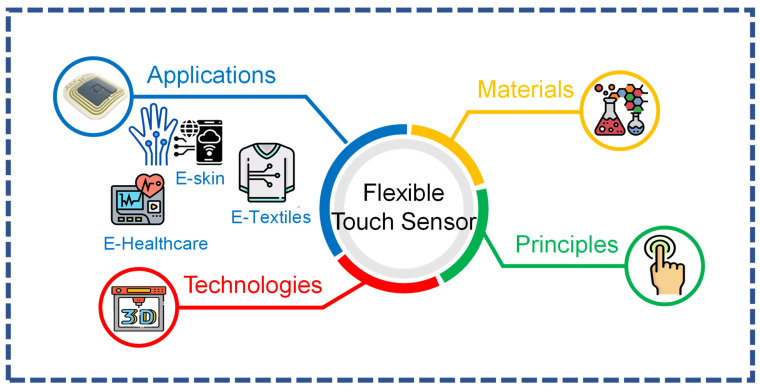
Overview of flexible touch sensors.

**Figure 4 sensors-22-04460-f004:**
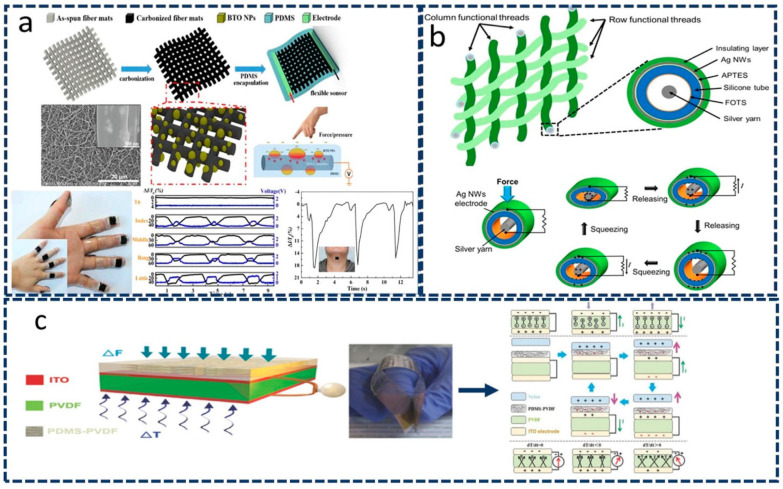
Three-principle integrated wearable sensors. (**a**) Carbonized electrospun nanofiber films composed of polyacrylonitrile/barium titanate (PAN-C/BTO) have been developed for the fabrication of multifunctional sensors. Reproduced with permission from the American Chemical Society (2018) [49]. (**b**) A power-generation sensor was obtained by integrating triboelectric, capacitive, and resistive mechanisms. Reproduced from Ref. [50]. (**c**) A flexible multifunctional sensor combining piezoelectric, triboelectric, and pyroelectric effects. Reproduced with permission from Wiley (2016) [51].

**Table 1 sensors-22-04460-t001:** Wearable technologies are classified according to their capabilities, properties, and application areas.

S.no.	Type	Capabilities	Properties	Applications
1.	Smartwatch	Displays information Navigation Fitness tracking Payment Communication	Voice- and touch-control interfaces Low operating power	Marketing, insurance Business, administration Education Professional sports, training Infotainment
2.	Fitness tracker	Activity tracking Physiological wellness Navigation Heart-rate monitor	High accuracy Lightweight Waterproof Wireless communication	Fitness Professional sport Healthcare Outdoor/indoor sports
3.	Smart eyewear	Visualization Communication Language interpretation Task coordination	Device is operated by touching the screen, speaking, shaking a hand, or moving one’s head Low operating power Provide direct access to sound	Surgery Logistics Aerospace and defense Infotainment Education
4.	Wearable camera	Live streaming Captures real-time photo and video	Smaller dimensions Night vision	Defense Education Industry Fitness
5.	Smart clothing	Tracking daily activities, heart rate, temperature, and body position Cooling or heating the body	No visual interaction with the user via display or screen Data are obtained by body sensors and actuators	Logistics Medicine Professional sports and fitness Military
6.	Wearable medical device	Physiological disorders Cardiovascular diseases Surgery Chronic diseases (e.g., diabetes) Dermatology Neuroscience Rehabilitation	Physiological tracking Pain management Sleep monitoring Glucose monitoring Brain-activity monitoring	Cardiovascular medicine Fitness Psychiatry Surgery Oncology Dermatology Respirology

**Table 2 sensors-22-04460-t002:** Summary of various sensors and sensing systems in terms of electrical and mechanical performance.

S.No.	Material	Principle	Response Time	Sensitivity (kPa^−1^)	Thickness	Working Range	Refs.
1.	Single-layer ions gel, copper electrode	Capacitive	<1 s	2.266	>200 µm	1–25 kPa	[16]
2.	PDMS, SWNTs, Si, PET	Resistive	20 ms	1	~900 µm	0.1–100 kPa	[12]
3.	Au/PET, PDMS	Capacitive	70 ms	0.42	100 µm	1–9 kPa	[1]
4.	Paper, CNTs, silver paste	Resistive	30 ms	2.56–5.67	>1 mm	0–20 kPa	[17]
5.	Microstructures, ITO-PET, PDMS, CNFs	Resistive	20–50 ms	3.6	>1.5 mm	0–2 kPa	[18]
6.	Hemispheric microstructures, Au, PDMS, Ag Paste	Resistive	26 ms	196	—	0–100 kPa	[19]
7.	Pyramid microstructures, PDMS, Au/Cr, PPy/PDMS,	Resistive	50 μs	>200	—	0.1–1000 Pa	[20]
8.	PDMS, silk fibroin, AgNWs,	Triboelectric	7 ms	—	0.12 mm	~90 V output, 8–22 kPa	[21]
9.	BCZT, PDMS particles	Piezoelectric	—	0.55	0.5 mm	28.8 V output, 50–1000 kPa	[22]
10.	Acrylate polymers, CNT, Ni-coated textiles	Multi-touch/resistive	24 ms	14.4		0–15 kPa	[23]
11.	Metal–insulator–metal (MIM), graphene field-effect transistor (GFET)	Piezoelectric	—	0.00455	—	23.54–94.18 kPa	[24]
12.	PDMS, graphite (pencil), paper	Multi-touch/capacitive	>200 ms	0.62	>250 µm	0.5–10 kPa	[25]

SWNTs, single-walled carbon nanotubes; PDMS, polydimethylsiloxane; CNTs, carbon nanotubes; CNFs, carbon nanofibers; ITO, indium tin oxide; PPy, polypyrrole; AgNWs, silver nanowires; BCZT, (Ba_0.85_Ca_0.15_)(Ti_0.90_Zr_0.10_)O_3_; PET, polyethylene terephthalate.

**Table 3 sensors-22-04460-t003:** Stretchable and flexible polymer/CNTs matrices’ sensor performances.

Types	Materials	Fabrication Process	Application	Gauge Factor	Strain (%)	Repeatability	Response Time	Refs
Filament strains sensors	MWCNT-TPU/SBS	Melt extrusion	Wearables and sports	GF = 26 for ε = 0–50%	~150%	Repeatable after the 5th cycle	~1 s	[77]
	Coaxial structure, ecoflex: sheath, core: MWCNT/ecoflex	Coaxial wet-spinning (CWS)	Expansion SHM and wearables	GF = 48 for ε < 50%	100%	Up to 3250 cycles	<1 s	[78]
	Acrylonitrile-butadiene styrene/MWCNT	Fused filament fabrication (FFF)	SHM	GF = 3.5 for ε = 3%	<4%	Fairly repeatable only after 40 cycles for 10 cycles	~1 s	[79]
	Coaxial structure, TPE: sheath, core: SWCNT	CWS	Wearables	GF = 1378 for ε = 330%	Up to 600%	Up to 10,000 cycles	<295 ms	[76]
Thin film-based strain sensor	MWNT/PDMS ecoflex/MWCNTs	Blending method	E-skin application	-	120–300%	-	-	[80]
	Polycarbonate-urethane (PCU)/ aligned CNT	Dry-spinnable MWCNT array by chloride-mediated CVD method	Wearable, real-time, human body motion sensing	10	500	180,000 cycles	15 ms	[81]
	Silicon lamina: dragon skin/CNTs	-	Human motion detection, prototypical data glove and respiration monitor	resolution < 1%	up to 300%	10,000 cycles at 100% strains	100 ms	[82]
	(poly-vinylpyrrolidone) PU/MWCNT laminate MWCNTs oxidized with KMnO4	Electrospinning	Human breath monitoring	450	300	1000 cycles	-	[83]
	0.48% CNTs modified by silane coupling agent (SCA)	Swelling/permeating method	Flexible sensor field	20	350	Not tested	-	[84]
	5.46 vol%. MWCNTs in OBC (elastomeric ethylene-α-octene block copolymer)	Melt mixing	Human motion detection	Aligned CNTs: 248 Random CNTs: 5.46	300	Not tested	-	[85]

## Data Availability

All data are contained within the article.

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
