# Peer review of "Recent Advances in Touch Sensors for Flexible Wearable Devices"

_sensors, 2022, doi:10.3390/s22124460_

Round 1
Reviewer 1 Report
Comment 1: Please, highlight the input of your own research groups in the appropriate sections.
Comment 2: Section 4, Author explained variety of nanofiller but lack of polymers. Especially, flexibility largely depends on polymer (matrix) in composites. Therefore, the variety of polymer could be added in this review paper.
Comment 3: Section 5, for better understanding of evaporation of heated source and condensation process, Figure should be added for reader’s understanding.
Reviewer 2 Report
Overall the authors have conducted a solid review of the recent advancements in the field of flexible touch sensors. I believe this paper can be further improved by addressing following minor comment and suggestions.
Comment - Last sentence on page 5: "These sensors are hypoallergenic..." - the authors compare flexible sensors to classical rigid electronic devices. The way this sentence is phrased the reader might have an impression that all flexible sensors are hypoallergenic - which is obviously not true. Some can be. Consider rephrasing.
This brings up the questions of practicality. If any wearable solution is to be adopted by the end user it should practical and safe to use, wear, wash, store, etc. The authors brush the subject of encapsulation in Section 6, however it would be interesting to expand this discussion. For example, when speaking of manufacturing processes, are they compatible with classical garment fabrication process (if we are speaking about e-textiles) or have to be introduced on the later stage via an labor intensive process? Are there studies of mechanical durability, longevity, wash-ability, etc. of the flexible textile sensors. Could certain sensors become potentially dangerous in case of mechanical failure (electrolyte spilling on the user's skin for instance)? And so on.
Reviewer 3 Report
Dear Authors,
In the introduction, please make sure that the market shares are listed in chronological order in the text for Figure 1; here, the reader gets a bit confused with the year numbers. The text should also mention what the rest is in each case.
You are writing: Wearables devices for human health have received intensive interest because they are an integral part of AI and the IoT. Isn't it just the other way around!?
In the case of restrictive pressure sensors, the pyramid structure is specifically discussed. Without studying the reference in detail, it is difficult to understand in this section exactly what the authors mean here. Please omit the exact structure information or explain it better.
Why is the multitouch principle placed in section 2.3? What is the connection to the heading piezo and triboelectric touch sensors?
You write: A wearable sensor based on a single-principle mechanism cannot detect multiple features. I don't think this is true: by cleverly interconnecting individual sensors of a measuring principle, it is very well possible to register different gestures or finger swipes, for example. AI in the background can also derive certain actions from the interconnection and measurement of several identical sensors.
To the Hybrid Multisensor Mechanisms:
Resitive pressure sensors (e.g., stitched or knitted pressure sensors) can also measure compression and tension or flexure.
"Various polymeric materials such as PVDF, polypropylene (PP), and other nanomaterials are required" Why is PP mentioned here? There is a vast number of other relevant polymer materials: e.g. PU, PET, ...
This selection is not clear to the reader here.
Figure 3 and Figure 4 show a selection of arbitrary examples. The division into one-, two-, and three-principle integration or active principle is unclear and difficult to understand. This should be clearly shown e.g. in a table.
"Triple-principle mechanisms involving piezoelectric, triboelectric, and pyroelectric effects...". What is meant here by pyroelectric?? Yet another principle? Here it goes quite confused, this must be simplified. Likewise, a nano generator is suddenly mentioned here, this has nothing to do with the topic.
To what extent is power generation included in the overview? That doesn't fit here, it's just a side issue.
The context of the individual independent lists of examples is unclear and poorly structured. A common thread should be shown here. Why is one technology mentioned but not another? The field is too diverse and broad to show a selection here.
Section 5 lacks all important textile surface formation methods, such as knitting, weaving, or even embroidery. Here, too, there are a large number of examples with which sensors can be built.
The article tries to summarize the very broad field of touch sensors in a review article. However, this has not been achieved, or only to some extent. A common thread needs to be shown and the examples need to be better organized. In addition, important textile sensor examples that serve as touch sensors are missing:e.g. weaving, knitting or embroidery. Textile layer structures are also completely missing.
Extensive research and restructuring of the examples is required.
Reviewer 4 Report
This manuscript reports Recent advances in touch sensors for flexible wearable devices. The topic is interesting and useful. It contains plenty of the recent progress of the touch sensors including the coupling effects. I recommend publishing with a minor revision. Here the reference is limited for this topic, the authors can cite more papers on the touch sensors, such as Journal of Materials Chemistry A, 2018, 6, 16548-16555. The author needs to double-check the format of section 5.
Round 2
Reviewer 3 Report
Thank you for considering all the proposed changes.
Now from my point of view o.k.